# The *GJB2* (Cx26) Gene Variants in Patients with Hearing Impairment in the Baikal Lake Region (Russia)

**DOI:** 10.3390/genes14051001

**Published:** 2023-04-28

**Authors:** Vera G. Pshennikova, Fedor M. Teryutin, Alexandra M. Cherdonova, Tuyara V. Borisova, Aisen V. Solovyev, Georgii P. Romanov, Igor V. Morozov, Alexander A. Bondar, Olga L. Posukh, Sardana A. Fedorova, Nikolay A. Barashkov

**Affiliations:** 1Laboratory of Molecular Genetics, Yakut Science Centre of Complex Medical Problems, Yaroslavskogo 6/3, 677019 Yakutsk, Russia; psennikovavera@mail.ru (V.G.P.); rest26@mail.ru (F.M.T.); sardaanafedorova@mail.ru (S.A.F.); 2Laboratory of Molecular Biology, Institute of Natural Sciences, M.K. Ammosov North-Eastern Federal University, Kulakovskogo 46, 677010 Yakutsk, Russia; cherdonovasasha96@gmail.com (A.M.C.); borisovatv96@gmail.com (T.V.B.); neloann@mail.ru (A.V.S.); gpromanov@gmail.com (G.P.R.); 3Novosibirsk State University, 630090 Novosibirsk, Russia; mor@niboch.nsc.ru (I.V.M.); posukh@bionet.nsc.ru (O.L.P.); 4Institute of Chemical Biology and Fundamental Medicine, Siberian Branch of the Russian Academy of Sciences, 630090 Novosibirsk, Russia; alex.bondar@mail.ru; 5Federal Research Center Institute of Cytology and Genetics, Siberian Branch of the Russian Academy of Sciences, 630090 Novosibirsk, Russia

**Keywords:** hearing impairment, *GJB2* gene, connexin 26 (Cx26), genotype-phenotype analysis, haplotype analysis, Baikal Lake region, Siberia, Russia

## Abstract

The *GJB2* (Cx26) gene pathogenic variants are associated with autosomal recessive deafness type 1A (DFNB1A, OMIM #220290). Direct sequencing of the *GJB2* gene among 165 hearing-impaired individuals living in the Baikal Lake region of Russia identified 14 allelic variants: pathogenic/likely pathogenic—nine variants, benign—three variants, unclassified—one variant, and one novel variant. The contribution of the *GJB2* gene variants to the etiology of hearing impairment (HI) in the total sample of patients was 15.8% (26 out of 165) and significantly differed in patients of different ethnicity (5.1% in Buryat patients and 28.9% in Russian patients). In patients with DFNB1A (n = 26), HIs were congenital/early onset (92.3%), symmetric (88.5%), sensorineural (100.0%), and variable in severity (moderate—11.6%, severe—26.9% or profound—61.5%). The reconstruction of the SNP haplotypes with three frequent *GJB2* pathogenic variants (c.-23+1G>A, c.35delG or c.235delC), in comparison with previously published data, supports a major role of the founder effect in the expansion of the c.-23+1G>A and c.35delG variants around the world. Comparative analysis of the haplotypes with c.235delC revealed one major haplotype G A C T (97.5%) in Eastern Asians (Chinese, Japanese and Korean patients) and two haplotypes, G A C T (71.4%) and G A C C (28.6%), in Northern Asians (Altaians, Buryats and Mongols). The variable structure of the c.235delC-haplotypes in Northern Asians requires more studies to expand our knowledge about the origin of this pathogenic variant.

## 1. Introduction

The *GJB2* (Cx26) gene (13q12.11, MIM 121011) pathogenic variants associated with autosomal recessive deafness type 1A (DFNB1A, OMIM #220290) are the main cause of hereditary non-syndromic hearing impairment (HI). The degree of hearing loss ranges from profound congenital deafness to mild, progressive HI manifesting in late childhood, and is highly dependent on certain *GJB2* genotypes [1]. At present, more than 560 allelic variants of the *GJB2* gene are known (https://www.ncbi.nlm.nih.gov/clinvar/?term=GJB2%5Bgene%5D, accessed on 20 March 2023). The average contribution of the *GJB2* pathogenic variants to the etiology of HI in the world was estimated to be 17.3%. The highest contribution of the *GJB2* pathogenic variants was shown in the countries of Europe (27.1% on average), and the lowest contribution was shown in countries located in sub-Saharan Africa (5.6% on average) [1]. Moreover, the frequency and spectrum of the *GJB2* pathogenic variants vary significantly between different regions of the world [1,2,3]. The heterogeneity of *GJB2* variants can be explained by different genetic effects related to evolutionary, demographical and social factors (genetic drift, founder effect, migrations, mutational hot spot, selective advantages, consanguineous and assortative marriages) [4,5,6,7,8,9,10,11,12,13,14,15,16,17,18,19,20,21,22,23,24,25,26,27,28].

The *GJB2* gene is one of the first candidate genes in the study of genetic etiology in patients with hearing loss, which is explained by its well-studied frequency and spectrum of variants in many countries. However, there are still many regions and populations where the contribution of this gene to HI remains unclear. The multiethnic population of Russia presents challenges for comprehensive population genetic studies due to the significant geographical distances and cultural differences between numerous ethnic groups living in Russia. The previous studies in Russia examined a cohort of 2569 patients with HI, where the total contribution of the *GJB2* pathogenic variants to the etiology of HI was 43%, with c.35delG being the most prevalent variant [29,30]. In the vast Siberian territory, the contribution of the *GJB2* pathogenic variants in the HI cases was previously described only for three administrative regions (the Altai Republic, the Tyva Republic and the Sakha Republic). The contribution of the *GJB2* gene to the etiology of HI was 15.1% among Altaian patients (the major variant being c.235delC), 22.3% in Tuvinian patients (the major variant being c.516G>C (p.Trp172Cys) and 51.2% among Yakut patients (the major variant being c.-23+1G>A) [31,32,33].

The aim of this study is molecular-genetic, genotype-phenotype and haplotype analyses of the *GJB2* gene among patients with HI, living in the Republic of Buryatia located in the Baikal Lake region (the Siberian part of Russia).

## 2. Materials and Methods

### 2.1. Patients

The DNA samples of 165 patients with HI from 160 unrelated families were collected in 2019. The majority of patients were of Buryat (47.8%; n = 79) and Russian ethnicity (46.1%; n = 76). Patients of other ethnicities accounted for 6.0% (n = 10). The males accounted for 41.2% (n = 68) and females—58.8% (n = 97). The average age was 50.7 ± 15.5 years (Table 1).

### 2.2. Clinical and Audiological Examinations

For each patient, a medical history was collected, including the information on previous illnesses, allergological history, injuries and/or surgeries, the use of ototoxic drugs and the exposure to industrial noise. The hearing thresholds were determined by pure-tone audiometry, using a clinical tonal audiometer “AA222” (“Interacoustics”, Middelfart, Denmark), according to the current clinical standards. Air-conduction and bone conduction thresholds were obtained at 0.125, 0.25, 0.5, 1, 2, 4 and 8 kHz. Severity of hearing loss was defined by pure tone average (PTA_0.5,1,2,4kHz_), as mild (25–40 dB), moderate (41–70 dB), severe (71–90 dB) or profound (above 90 dB). The median hearing thresholds in PTA_0.5,1,2,4kHz_ of *GJB2* genotypes were compared using the Mann–Whitney U-test. Differences were considered statistically significant at *p* < 0.05.

### 2.3. GJB2 Gene Sequence Analysis

DNA was extracted using the phenol-chloroform method from the blood leukocytes. Amplification of coding (exon 2), non-coding (exon 1) and flanking intronic regions of the *GJB2* gene was performed by PCR on a T100 thermocycler (Bio-Rad, Hercules, NY, USA) using the following primers: 5′-CCGGGAAGCTCTGAGGAC-3′ and 5′-GCAACCGCTCTGGGTCTC-3′ for amplification of exon 1 [34]; and 5′-TCGGCCCCAGTGGTACAG-3′ and 5′-CTGGGCAATGCGTTAAACTGG-3′ for amplification of exon 2 [35,36,37]. The PCR products were subjected to Sanger sequencing, using the same primers on ABI PRISM 3130XL (Applied Biosystems, Waltham, MA, USA) at the Genomics Core Facility of Institute of Chemical Biology and Fundamental Medicine, Siberian Branch of the Russian Academy of Sciences (Novosibirsk, Russia). DNA sequence variations were identified by comparison with the *GJB2* gene reference sequences: chr13 (GRCh38.p13), NC_000013.11, NG_008358.1, NM_004004.6 and NP_003995.2 (NCBI, Gene ID: 2706).

### 2.4. Screening of Large DFNB1 Deletions

Screening of the large DFNB1 deletions was performed using oligonucleotide primers for the detection of break-point junction fragment specific for 309 kb—del(*GJB6*-D13S1830)—*GJB6* F5′-TTTAGGGCATGATTGGGGTGATTT-3′ and R5′-CACCATGCGTAGCCTTAACCATTT-3′ [38], for 232 kb—del(*GJB6*-D13S1854) F5′-TCATAGTGAAGAACTCGATGCTGTTT-3′ and R5′-CAGCGGCTACCCTAGTTGTGGTT-3′ [38] with internal control fragment (*GJB6*, exon 1) F5’-CGTCTTTGGGGGTGTTGCTT-3’ and R5’-CATGAAGAGGGCGTACAAGTTAGAA-3’ (*GJB6*, exon 1). Screening of the 101 kb—del(*GJB2*-d13S175) was performed using oligonucleotide primers for the detection of break-point junction fragment F5′-GCTCTGCCCAGATGAAGATCTC-3′, R5′-CCTTCCAGGAGAGTTCACAACTC-3′ with internal control fragment F5′-GTGATTCCTGTGTTGTGTGCATTC-3′, R5′-CCTCATCCCTCTCATGCTGTC-3′ (*GJB2*, exon 2) [29].

### 2.5. In Silico Pathogenicity Analysis

Analysis of the allelic variants with uncertain clinical significance (VUS) located in non-coding upstream 5′ region of the *GJB2* gene was performed using the Ensemble centroid instrument generated in the Sfold software 2.2. (https://sfold.wadsworth.org/cgi-bin/srna.pl, accessed on 15 September 2022).

### 2.6. Haplotypes Analysis

For the haplotype analysis of chromosomes with c.-23+1G>A (n = 2 chromosomes), c.35delG (n = 28 chromosomes) and c.235delC (n = 2 chromosomes), we performed genotyping of 12 SNP markers (rs1932429, rs5030701, rs7987144, rs7994748, rs2274084, rs2274083, rs3751385, rs5030700, rs11841024, rs2313477, rs747931 and rs2031282), flanking the region of 84,536 kB in the q12.11–q12.12 locus of chromosome 13. Primer sequences for genotyping of 12 SNP markers are presented in Appendix A. For comparative haplotype analyses of chromosomes with c.-23+1G>A, c.35delG or c.235delC, we summarized our data and the data on appropriate SNPs genotyping from previously published studies [6,7,14,21,39], in each case using analyzed SNPs overlapping with our study.

### 2.7. Brief Information about Studied Region

The Republic of Buryatia includes 21 districts and two cities (Ulan-Ude and Severobaikalsk) (https://egov-buryatia.ru, accessed on 15 September 2022), with an area of 351.3 thousand km^2^. This region of the Russian Federation borders with Mongolia. The population of the Republic of Buryatia is 978,600 people, with an average density of 2.78 people/km^2^. The major ethnic groups are Buryats (30.1%) and Russians (59.4%) (https://burstat.gks.ru/vpn2020, accessed on 25 January 2023). The Buryats are a Mongolic-speaking people and one of the largest indigenous groups in Siberia. Buryats share many customs with other Mongols, including nomadic herding and using portable dwellings—yurts. The majority of the Buryat population lives in the Republic of Buryatia, Irkutsk Oblast’ and Zabaykalsky Krai of Russia. Buryats also live in the northeastern part of Mongolia and China (Inner Mongolia).

### 2.8. Ethical Control

All patients gave written informed consent for participation in the study. This study was approved by the local Biomedical Ethics Committee at the Yakut Scientific Center of Complex Medical Problems, Yakutsk, Russia (Yakutsk, protocol No. 50 of 24 December 2019).

## 3. Results

### 3.1. Identified Variants in the GJB2 Gene

Large DFNB1 deletions of 309 kb—del(*GJB6*-D13S1830), 232 kb—del(*GJB6*-D13S1854) and 101 kb—del(*GJB2*-d13S175) were not found in our sample. Sequencing of the coding (exon 2) and non-coding (exon 1) with flanking intronic regions of the *GJB2* gene in 165 patients with HI from the Republic of Buryatia revealed fourteen allelic variants. In the protein-coding region (exon 2) of the *GJB2* gene, eleven variants were identified. In the non-coding regions (intron 1 and 5′UTR region) three variants were identified (Figure 1). Nine allelic variants are known as pathogenic or likely pathogenic (PLP), three variants are known as benign, one variant c.-254C>T is unclassified (uncertain significance) and one variant c.-49G>A was not described previously (Figure 1).

Analysis of allelic variants with uncertain clinical significance (c.-49G>A and c.-254C>T) located in the 5′ non-coding region of *GJB2* was performed using Sfold software 2.2. (https://sfold.wadsworth.org/cgi-bin/srna.pl accessed on 25 January 2023) (Appendix A). The in silico prediction of mRNA folding revealed no significant differences in conformation changes between normal sequences and sequences with c.-49G>A and c.-254C>T variants. These variants do not impact on any known functional motifs (TATA and GC boxes) (Figure 1) and most likely do not have clinical significance.

### 3.2. GJB2 Genotypes in Patients with HI

Eighteen different *GJB2* genotypes were identified in patients with HI in the Republic of Buryatia (n = 165). Among them, seven *GJB2* genotypes with biallelic pathogenic/likely pathogenic (PLP) variants in compound heterozygous and in homozygous state were found in 26 patients (15.8%) (Table 2).

Six different *GJB2* genotypes with single recessive pathogenic variant were found in nine (5.5%) patients. Five different *GJB2* genotypes with benign and unclassified variants were detected in 32 (19.4%) patients. In the other 98 (59.4%) patients we found no changes in the *GJB2* gene sequence (Table 2). Two heterozygous substitutions c.[-254C>T(;)516G>C] and c.[79G>A(;)341A>G] are presumably in the *cis*-position, which was previously shown in other studies [31,32,33,40,41]. In one patient with heterozygous c.23C>T p.(Thr8Met) variant, a homozygous c.457G>A p.(Val153Ile) variant was also found (Table 2). Based on clinical significance of the identified *GJB2* variants, the contribution of biallelic PLP variants to the etiology of HI in the total sample of patients in our study was 15.8% (26/165) (Table 2).

### 3.3. Contribution of the GJB2 Variants to the Etiology of HI in Buryat and Russian Patients

Furthermore, we subdivided our sample of HI patients into two main groups based on their ethnicity (Figure 2). Contribution of the *GJB2* variants to the etiology of HI among Buryat patients was lower—5.1% (4/79), compared to Russian patients—28.9% (22/76).

### 3.4. Audiological Analysis in Patients with GJB2 Gene Variants

In 26 patients with the *GJB2* variants in a homozygous or compound heterozygous state, hearing loss is characterized as early (detected up to 3 years) (92.3%), symmetric (88.5%), sensorineural (100.0%) and variable in severity (moderate—11.6%, severe—26.9%, and profound—61.5%). For detailed audiological analyses we divided these 26 patients into seven different biallelic *GJB2* genotypes: c.[35delG];[35delG] (n = 16), c.[-23+1G>A];[35delG] (n = 5), c.[-23+1G>A];[327_328delGGinsA] (n = 1), c.[35delC];[299_300delAT] (n = 1), c.[235delC];[235delC] (n = 1), c.[-23+1G>A];[-23+1G>A] (n = 1), c.[-23+1G>A];[-254C>T(;)516G>C] (n = 1). The most common *GJB2* genotype c.[35delG];[35delG] (n = 16 individuals, 32 ears) was chosen as a reference. We found that audiological profiles of the c.[-23+1G>A];[35delG], c.[-23+1G>A];[327_328delGGinsA], c.[35delC];[299_300delAT], and c.[235delC];[235delC] genotypes have a relatively gentle curve and were similar with the reference genotype c.[35delG];[35delG]. However, two *GJB2* genotypes c.[-23+1G>A];[-23+1G>A] and c.[-23+1G>A];[-254C>T(;)516G>C] demonstrated the flat curve with better preservation of hearing thresholds in the audiological profiles (Figure 3).

For the analysis of the hearing features in the speech frequencies, the median hearing thresholds in the PTA_0.5,1.0,2.0,4.0kHz_ of six *GJB2* genotypes were compared in pairs with the reference group by the Mann–Whitney U-test. The PTA_0.5,1.0,2.0,4.0kHz_ of hearing threshold in patients with the reference *GJB2* genotype c.[35delG];[35delG] (median 102 dB, profound HI) was comparable to the *GJB2* genotypes: c.[-23+1G>A];[35delG] (median 106.2 dB, profound HI), c.[-23+1G>A];[327_328delGGinsA] (median 96.8 dB, profound HI), c.[35delC];[299_300delAT] (median 92.5 dB, profound HI), and c.[235delC];[235delC] (median 117.5 dB, profound HI) (*p* > 0.05). Two *GJB2* genotypes c. [-23+1G>A];[-23+1G>A] (median 66.8 dB, moderate HI) and c.[-23+1G>A];[-254C>T(;)516G>C] (median 64.3 dB, moderate HI) demonstrated significantly better hearing thresholds, compared to the reference *GJB2* genotype (*p* < 0.05) (Figure 4).

### 3.5. Genotypes of 12 SNP Markers Flanking the GJB2 Gene

For haplotype analysis, we genotyped 12 SNPs (rs1932429, rs5030701, rs7987144, rs7994748, rs2274084, rs2274083, rs3751385, rs5030700, rs11841024, rs2313477, rs747931 and rs2031282, further designated as 1, 2, 3, 4, 5, 6, 7, 8, 9, 10, 11 and 12, respectively) in one Buryat patient homozygous for c.-23+1G>A (2 chromosomes), in 14 Russian patients homozygous for c.35delG (28 chromosomes) and in one Buryat homozygous for c.235delC (2 chromosomes). The genotypes of 12 SNP markers flanking the *GJB2* gene in 16 homozygous patients are presented in Figure 5.

## 4. Discussion

In the present study, molecular genetic analysis of the non-coding (exon 1) and coding (exon 2) regions of the *GJB2* gene was performed on 165 individuals with HI from the Republic of Buryatia (Eastern Siberia, Russia). Among fourteen identified allelic variants, nine variants are known as pathogenic or likely pathogenic, three variants are known as benign, one variant (c.-254C>T) is unclassified and one variant (c.-49G>A) has not been previously described. The total contribution of the *GJB2* variants to the etiology of HI in observed patients in the Republic of Buryatia was 15.8% (26 out of 165); however, there were significant differences between two main ethnic groups of patients: 28.9% in Russians and 5.1% in Buryats. The contribution of the *GJB2* variants in Russian patients defined in this study (28.9%) corresponds to the data on Russian patients with HI (33.0%) living in neighboring Siberian region (the Sakha Republic) [32]. However, the low contribution of the *GJB2* variants to HI in Buryat patients (5.1%) belonging to one of the indigenous Siberian populations, is lower compared to other Siberian ethnic groups (15.1%—in Altaian patients, 22.3%—in Tuvinian patients, and 51.2%—in Yakut patients) [31,32,33,42], while it is similar to the proportion of *GJB2*-related HI in patients from neighboring Mongolia (4.5–6.9%) [14,43]. It should be noted that the frequency of HI associated *GJB2* variants in Mongolia is one of the lowest in the world [1].

Audiological analysis in 26 patients with the *GJB2* variants revealed that HI was congenital/early onset (92.3%), symmetric (88.5%), sensorineural (100.0%) and variable in severity (moderate—11.6%, severe—26.9% and profound—61.5%). The clinical effect of the *GJB2*-genotype c.[35delG];[35delG] 108 dB in PTA, profound HI) was previously demonstrated by the multicenter genotype-phenotype studies [44,45]. In this regard, the *GJB2* genotype c.[35delG];[35delG] was used as a reference (in our study—median 102 dB in PTA, profound). The detailed analysis of severity in PTA_0.5,1.0,2.0,4.0kHz_ demonstrated better hearing thresholds for the c.[-23+1G>A];[-23+1G>A] genotype (median 66.8 dB, moderate HI) and for the c.[-23+1G>A];[-254C>T(;)516G>C] genotype (median 64.3 dB, moderate HI) (*p =* 0.022), compared to the c.[35delG];[35delG] genotype (median 102 dB, profound HI) (Figure 4). To our knowledge, our earlier study [42] is the only audiological analysis of a large cohort of patients (n = 40) homozygous for the c.-23+1G>A variant. In that study we reported the median PTA_0.5,1.0,2.0,4.0kHz_ to be 86 dB (severe HI), which is better than that in patients with c.[35delG];[35delG] genotype (102 dB, profound HI) (Figure 4). Taken together, our findings in patients with combined genotypes c.[-23+1G>A];[-254C>T(;)516G>C] (median 64.3 dB, moderate HI) indicate that these *GJB2*-variants (c.-23+1G>A and c.516G>C) may be associated with more mild phenotypes than observed with homozygous c.35delG genotypes. It is possible that our sample of patients is biased towards severe and profound HI, and as a result, we did not find more cases with milder phenotypes.

Interestingly, despite the relatively small total contribution of the *GJB2* variants in our sample, we found three different types of homozygotes for c.-23+1G>A, c.35delG or c.235delC. Based on genotyping of 12 SNP markers in chromosomes with these three variants (Figure 5), we performed a haplotype analysis in comparison with previously published data using overlapping SNPs (the set of the same SNP markers used in the present and previous studies) [6,7,14,21,39]. This allowed us to reconstruct mutant haplotypes in 254 chromosomes with c.-23+1G>A by six overlapping SNPs (SNPs 1, 3, 5, 6, 9 and 10) (Figure 5). The common haplotype C C G A A C (99.2%) was found among all examined patients homozygous for c.-23+1G>A from Turkey, Mongolia and from European and Siberian parts of Russia (Appendix A, Figure 6). This finding supports the hypothesis about the common origin of c.-23+1G>A variant [27,39]. Reconstruction of the mutant haplotypes by four overlapping SNPs (SNPs 7, 8, 11, 12) (Figure 5) in 256 chromosomes with c.35delG revealed six different c.35delG-haplotypes. Among them, the T C T C haplotype had the highest frequency among all mutant chromosomes (60.9%) (Appendix A, Figure 6), which also supported the hypothesis of a single origin of this pathogenic variant [6,43,46,47,48,49,50,51,52,53,54,55,56,57].

The most interesting results were obtained from the reconstruction of 54 mutant chromosomes with c.235delC using four overlapping SNPs (SNPs 5, 6, 7 and 11) (Figure 5). Three different c.235delC-haplotypes (G A C T, G A T T and G A C C) were reconstructed. In this regard, we divided the Asian patients with c.235delC in two groups: the Northern Asians (Altaian, Buryat and Mongolian patients) (this study and [7,21]) and the Eastern Asians (Chinese, Korean and Japanese patients) [7] (Figure 6). The G A T T haplotype was previously found in Eastern Asians with low frequency (8.3%, only in one chromosome of Japanese patients) (Appendix A, Figure 6). The c.235delC-haplotype G A C T was presented with high frequency in the Eastern Asians (97.5%, CI: 87.1–99.4%) and with lower frequency in the Northern Asians (71.4%, CI: 44.9–88.2%) (Appendix A). The c.235delC-haplotype G A C C was absent in Eastern Asians (0%, CI: 0–0.8%) and was present only in Northern Asians (28.6%, CI: 11.8–55.1%) (*p <* 0.05) (Figure 6, Appendix A). Two main haplotypes, G A C T and G A C C, differ by one distant SNP marker (rs747931, ~62.5 kB from c.235delC, underlined).

The diversity of the c.235delC haplotypes in different geographical regions may indicate an independent origin of this *GJB2* pathogenic variant in Eastern and Northern Asians. However, all previous research focusing on the analysis of the haplotypes bearing c.235delC suggested the hypothesis on the single origin of c.235delC [7,20,21,23,43,58,59,60]. Thus, in the study by Yan et al. (2003), seven SNPs flanking c.235delC were analyzed in patients with HI from China, Japan, Korea and Mongolia, and the only one haplotype A G A C (SNP2-V27I-E114G-SNP1) associated with c.235delC was found [7]. In addition, the age of c.235delC (~11,500 years) estimated by one distant SNP marker (SNP6 or rs747931) was calculated [7]. It is worth noting that this particular SNP differentiated the haplotypes found in our study (G A C T and G A C C, underlined) (Appendix A, Figure 6). Assuming a probable recombination of this distant SNP (rs747931) and a potential effect of genetic drift, our data on the structure of variable haplotypes in North Asians may be consistent with the hypothesis proposed by Yan et al. (2003) about a single origin of c.235delC in the Baikal Lake region and its spreading to Mongolia, China, Korea and Japan through subsequent migrations [7]. However, in recent studies, a “younger” age of c.235delC was determined: ~6500 years and ~1125–3150 years in the Japanese and Altaian carriers of c.235delC, respectively [20,21]. These discrepancies can be explained by differences in the methods of the age estimation, the panels of used genetic markers, and the sample sizes in different studies [20,21], although this may be due to different genetic background of populations studied. We have identified the c.235delC haplotype for the first time in the presumed region of c.235delC origin (the Baikal Lake region) and believe that subsequent studies of the structure of haplotypes bearing c.235delC will elucidate the origin of this *GJB2* pathogenic variant.

## 5. Conclusions

In this study, the comprehensive analysis of the *GJB2* gene was performed among patients with HI of different ethnicities living in the Baikal Lake region in the Siberian part of Russia. We found that the contribution of the *GJB2* gene variants to the etiology of hearing loss in Buryat patients is significantly lower (5.1%) than in Russian patients (28.9%). Further extensive studies (including the NGS technology) are necessary to elucidate the unidentified genetic causes of hearing loss in patients examined in our study. Audiological analysis revealed that the genotype with c.-23+1G>A and c.516G>C variants may be associated with a milder phenotype than the genotypes homozygous for the c.35delG variant. The haplotype analysis in chromosomes with c.-23+1G>A and c.35delG performed in comparison with available published data supports a major role of the founder effect in the expansion of these variants around the world. The diversity of the c.235delC haplotypes in Northern Asians may indicate an independent as well as the common origin of this *GJB2* pathogenic variant in Asia. We believe that subsequent studies of the structure of haplotypes bearing c.235delC will elucidate the origin of this *GJB2* pathogenic variant.

## Figures and Tables

**Figure 1 genes-14-01001-f001:**
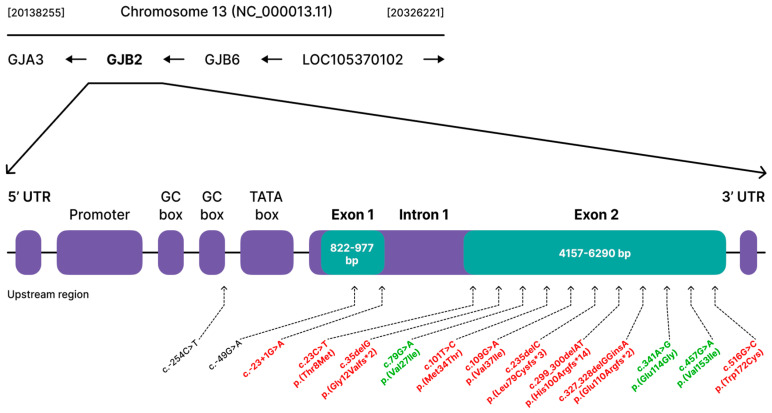
Identified allelic variants of the *GJB2* gene among 165 patients with HI in the Republic of Buryatia. Schematic structure of the *GJB2* gene is based on NC_000013.11 reference sequence (https://www.ncbi.nlm.nih.gov/gene/2706 accessed on 25 January 2023); Pathogenic/likely pathogenic variants are shown in red, benign variants are shown in green and variants with uncertain significance (including the novel c.-49G>A variant) are shown in black.

**Figure 2 genes-14-01001-f002:**
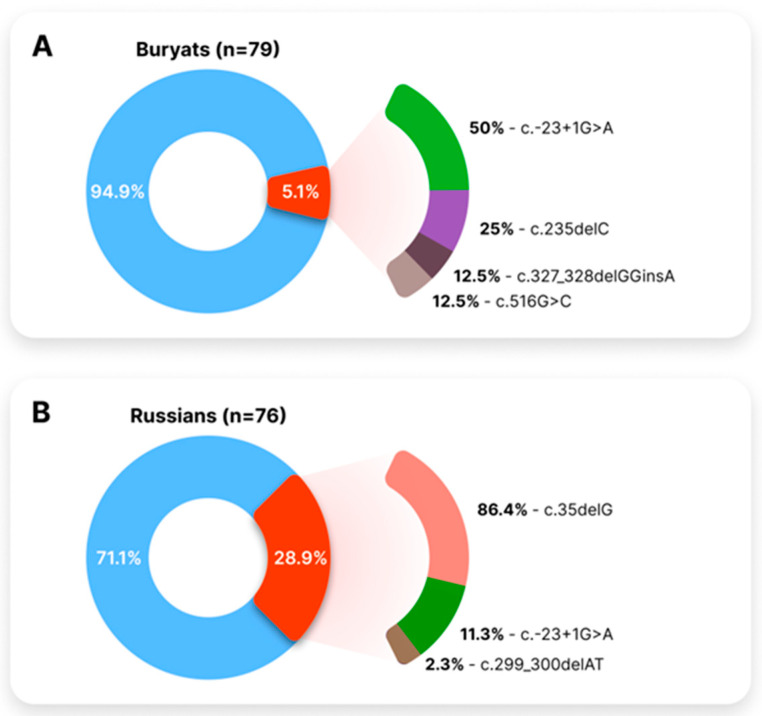
Contribution of the *GJB2* variants to the etiology of HI in studied patients. (**A**)—Contribution of the *GJB2* variants to the etiology of HI in Buryat patients. (**B**)—Contribution of the *GJB2* variants to the etiology of HI in Russian patients. Proportions of *GJB2*-negative and *GJB2*-positive patients are shown in blue and red, accordingly. The allele frequency of the *GJB2* variants was calculated in unrelated patients.

**Figure 3 genes-14-01001-f003:**
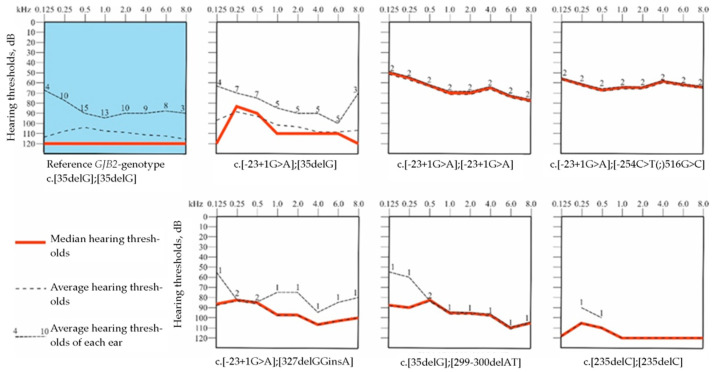
Audiological profiles in patients with the *GJB2* gene variants. The reference group (shown in blue)—patients with genotype c.[35delG];[35delG] (16 individuals, 32 ears).

**Figure 4 genes-14-01001-f004:**
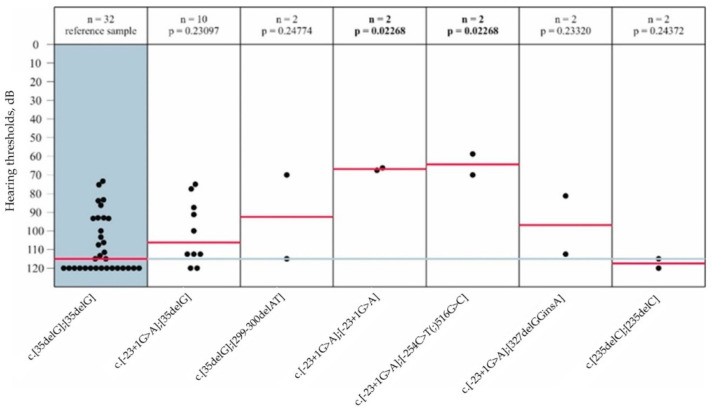
The PTA_0.5,1.0,2.0,4.0kHz_ hearing thresholds in patients with HI caused by the *GJB2* gene variants. Reference group (shown in blue)—patients with the c.[35delG];[35delG] genotype (16 individuals, 32 ears). The red lines—the median hearing thresholds in the PTA_0.5,1.0,2.0,4.0kHz._ The blue line—the median hearing threshold in the PTA_0.5,1.0,2.0,4.0kHz_ of the reference group. Statistically significant differences (*p* < 0.05) are shown in bold.

**Figure 5 genes-14-01001-f005:**
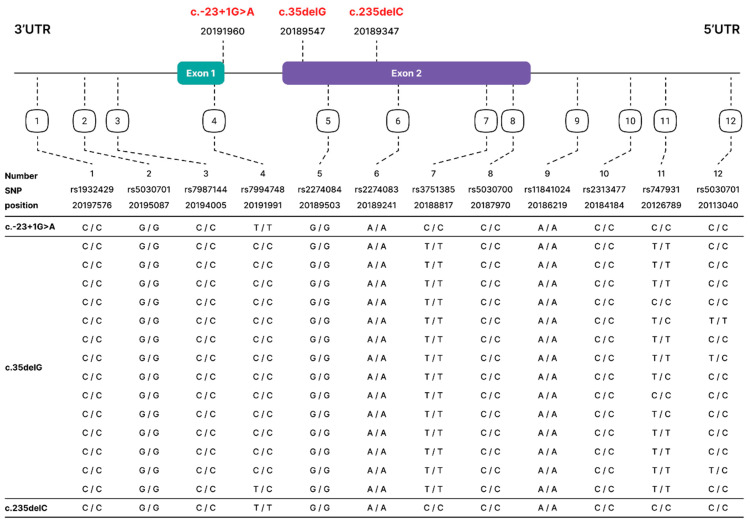
The genotypes of 12 SNPs, flanking the *GJB2* gene in 16 patients homozygous for c.-23+1G>A (n = 1), c.35delG (n = 14), and c.235delC (n = 1). Positions of c.-23+1G>A, c.35delG, c.235delC and 12 SNPs were defined according to GRCh38.p13 Genome Assembly (https://www.ncbi.nlm.nih.gov/assembl, accessed on 25 January 2023).

**Figure 6 genes-14-01001-f006:**
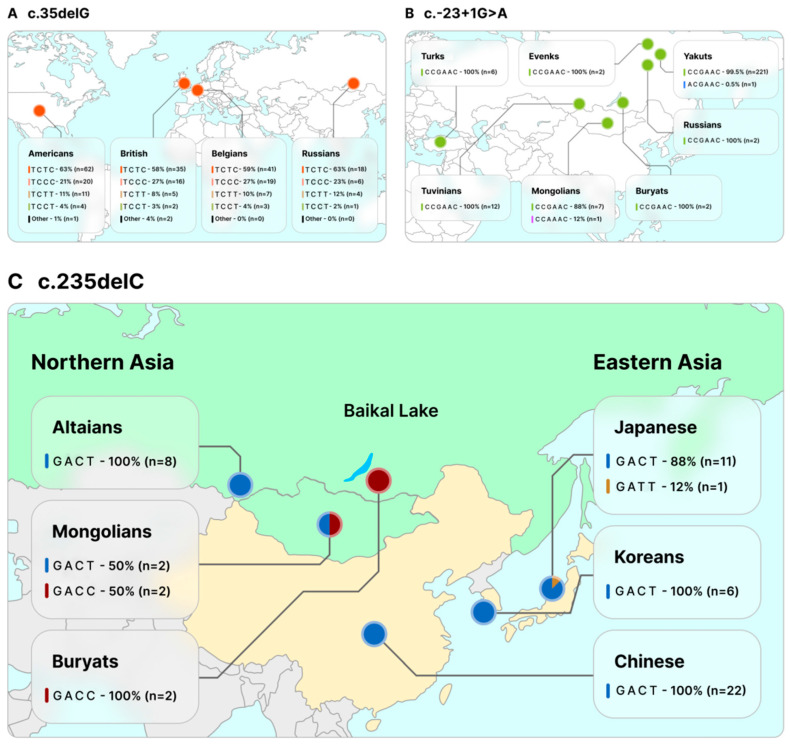
The diversity of the haplotypes with three pathogenic *GJB2* variants. (**A**) The haplotypes with c.35delG. (**B**) The haplotypes with c.-23+1G>A. (**C**) The haplotypes with c.235delC.

**Table 1 genes-14-01001-t001:** Patients with HI in the Republic of Buryatia.

Sample	n (%)	Males	Females	Age
Buryats	79 (47.8%)	31 (39.2%)	48 (60.8%)	46.9 ± 14.3
Russians	76 (46.1%)	32 (42.1%)	44 (57.9%)	54.8 ± 15.7
Others	10 (6.1%)	5 (50.0%)	5 (50.0%)	49.2 ± 15.6
Total	165 (100%)	68 (41.2%)	97 (58.8%)	50.7 ± 15.5

n—number of patients.

**Table 2 genes-14-01001-t002:** *GJB2* genotypes among 165 patients with HI in the Republic of Buryatia.

#	*GJB2*-Genotypes	Buryats		Russians		Others		Total	
		n = 79	F (%)	n = 76	F (%)	n = 10	F (%)	n = 165	F (%)
*GJB2* genotypes with biallelic PLP variants									
1	c.[-23+1G>A];[-23+1G>A]	1	1.2	-	-	-	-	1	0.6
2	c.[-23+1G>A];[35delG]	-	-	5	6.6	-	-	5	3.0
3	c.[-23+1G>A];[327_328delGGinsA]	1	1.2	-	-	-	-	1	0.6
4	c.[-23+1G>A];[-254C>T(;)516G>C]	1	1.2	-	-	-	-	1	0.6
5	c.[35delG];[35delG]	-	-	16	21.1	-	-	16	9.7
6	c.[35delG];[299_300delAT]	-	-	1	1.3	-	-	1	0.6
7	c.[235delC];[235delC]	1	1.2	-	-	-	-	1	0.6
Total		4	5.1	22	28.9	-	-	26	15.8
*GJB2* genotypes with monoallelic PLP variants									
8	c.[-23+1G>A];[wt]	1	1.2	1	1.3	-	-	2	1.2
9	c.[23C>T(;)457G>A];[457G>A]	-	-	1	1.3	-	-	1	0.6
10	c.[35delG];[wt]	-	-	1	1.3	-	-	1	0.6
11	c.[101T>C];[wt]	-	-	1	1.3	-	-	1	0.6
12	c.[109G>A];[wt]	-	-	2	2.6	-	-	2	1.2
13	c.[109G>A];[79G>A]	1	1.2	1	1.3	-	-	2	1.2
Total		2	2.5	7	9.2	-	-	9	5.5
*GJB2* genotypes with benign variants									
14	c.[-49G>A];[wt]	-	-	1	1.3	-	-	1	0.6
15	c.[79G>A];[wt]	18	22.8	1	1.3	1	10	20	12.1
16	c.[79G>A(;)341A>G];[wt]	5	6.3	2	2.6	2	20	9	5.5
17	c.[79G>A(;)341A>G];[79G>A]	1	1.2	-	-	-	-	1	0.6
18	c.[457G>A];[wt]	-	-	1	1.3	-	-	1	0.6
Total		24	30.4	5	6.6	3	30	32	19.4
*GJB2* genotype without sequence changes (wild type)									
c.[wt];[wt]		49	62.0	42	55.3	7	70	98	59.4

F—genotype frequency, n—number of individuals with HI; and PLP variants—pathogenic/likely pathogenic variants, # number.

## Data Availability

The data presented in this study are available on request from the corresponding author.

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
