# Peer review of "The GJB2 (Cx26) Gene Variants in Patients with Hearing Impairment in the Baikal Lake Region (Russia)"

_genes, 2023, doi:10.3390/genes14051001_

Round 1
Reviewer 1 Report
The manuscript by Pshennikova et al. describes the identification of GJB2 variants in a cohort of 165 deaf individuals from the Lake Baikal region of Russia. The authors show that GJB2 pathogenic variants underlie the hearing impairment of nearly 16$ of patients in the cohort, with different proportions among ethnic Russians and Buryats. Haplotype analysis of patients indicate a common founder effect for two of the most prevalente variants (c.35delG and c.-23+1G>A) and two different haplotypes for the variant c.235delC, indicating differnet possibilities for the origin of this variant in Asian populations.
COMMENTS
(1) In silico pathogenicity analysis of c.-254C>T and c.-49G>A (lines 172-175). This results should be better presented. The authors should indicate that they just performed an in silico prediction of mRNA folding and saw no significant difference between wt and variant. They should additionally argue that since the variants do not impact on any known functional motifs (e.g. TATA o GC boxes, etc) and there seems to be no misfolding, the variants are unlikely to be pathogenic.
(2) Proper naming of variants according to HGVS nomenclature rules (lines 157-171). Please use Mutalyzer software (http://mutalyzer.nl) to present correct nomenclature of the frameshift variants. For instance, c.35delG at the protein level becomes p.Gly12Valfs*2 and not just p.Gly12Valfs.
line 198: Typo: "varaints" should be "variants".
line 316: "overlapping SNPs". SNPs are point variantioons and hence cannot be overlapping... Do the authors mean that those SNPs overlap the GJB2 gene?
(3) Discussion, lines 279-296. The authors should discuss the possibility that their samples is biased towards higher hearing losses due to the possibility that mildly affected individuals are not aware of their heaaring loss or do not seek consultation with a physician.
(4) Discussion, lines 315-354. Since the only difference between the two major haplotypes of c.235delC happens at the outermost SNP, the authors should discuss the possibility of a single origin and genetic drift and recombination being responsible for the observed difference.
Author Response
Responses to Reviewer 1
Dear Reviewer,
On behalf of the authors, I would like to thank you for your efforts in reviewing our manuscript, and I greatly appreciate your comments.
The manuscript by Pshennikova et al. describes the identification of GJB2 variants in a cohort of 165 deaf individuals from the Lake Baikal region of Russia. The authors show that GJB2 pathogenic variants underlie the hearing impairment of nearly 16$ of patients in the cohort, with different proportions among ethnic Russians and Buryats. Haplotype analysis of patients indicate a common founder effect for two of the most prevalente variants (c.35delG and c.-23+1G>A) and two different haplotypes for the variant c.235delC, indicating differnet possibilities for the origin of this variant in Asian populations.
COMMENTS
(1) In silico pathogenicity analysis of c.-254C>T and c.-49G>A (lines 172-175). This results should be better presented. The authors should indicate that they just performed an in silico prediction of mRNA folding and saw no significant difference between wt and variant. They should additionally argue that since the variants do not impact on any known functional motifs (e.g. TATA o GC boxes, etc) and there seems to be no misfolding, the variants are unlikely to be pathogenic.
- In accordance with this comment, we have added the following new phrases about in silico analysis in the "Results" section (Lines: 173-180):
“Analysis of allelic variants with uncertain clinical significance (c.-49G>A and c.-254C>T) located in the 5’ non-coding region of GJB2 was performed using Sfold software (https://sfold.wadsworth.org/cgi-bin/srna.pl) (Supplementary Materials, Chapter 2). The in silico prediction of mRNA folding revealed no significant differences in conformation changes between normal sequences and sequences with c.-49G>A and c.-254C>T variants. These variants do not impact on any known functional motifs (TATA and GC boxes) (Figure 1) and most likely do not have clinical significance.”
(2) Proper naming of variants according to HGVS nomenclature rules (lines 157-171). Please use Mutalyzer software (http://mutalyzer.nl) to present correct nomenclature of the frameshift variants. For instance, c.35delG at the protein level becomes p.Gly12Valfs*2 and not just p.Gly12Valfs.
-We changed the nomenclature of the frameshift variants, according to HGVS rules (Lines 161-163 and 168-170).
line 198: Typo: "varaints" should be "variants".
- This typo is fixed (Line: 198).
line 316: "overlapping SNPs". SNPs are point variantioons and hence cannot be overlapping... Do the authors mean that those SNPs overlap the GJB2 gene?
- We have used the term "overlapping" to refer to the set of the same SNP markers used in the present and previous studies. We added this comment in the first sentence where we used this term (Lines: 305-306):
“Based on genotyping of 12 SNP markers in chromosomes with these three variants (Figure 5), we performed a haplotype analysis in comparison with previously published data using overlapping SNPs (the set of the same SNP markers used in the present and previous studies) [6, 7, 14, 21, 39].”
(3) Discussion, lines 279-296. The authors should discuss the possibility that their samples is biased towards higher hearing losses due to the possibility that mildly affected individuals are not aware of their hearing loss or do not seek consultation with a physician.
- We agree with this comment and added this assumption in the Discussion section (Lines: 298-299):
“It is possible that our sample of patients is biased towards severe and profound HI, and as a result, we did not find more cases with milder phenotypes.”
(4) Discussion, lines 315-354. Since the only difference between the two major haplotypes of c.235delC happens at the outermost SNP, the authors should discuss the possibility of a single origin and genetic drift and recombination being responsible for the observed difference.
- We have added the following phrase to the Discussion section (Lines: 344-348):
“Assuming a probable recombination of this distant SNP (rs747931) and a potential effect of genetic drift, our data on the structure of variable haplotypes in North Asians may be consistent with the hypothesis proposed by Yan et al. (2003) about a single origin of c.235delC in the Baikal Lake region and its spreading to Mongolia, China, Korea and Japan through subsequent migrations [7].”

Reviewer 2 Report
This is a well-written and well-presented paper. I suggest a few minor grammatical changes:
(deleted words have strikethough, inserted words are underlined)
Line 60: “… Russia presents the challenges for …” (i.e. omit “the”)
Line 61: “… singificant geographical distances and culturual differences between among numerous …”
Line 86: “ and contact with exposure to industrial noise.”
Line 124: The phrase “Ensemble centroid instrument” is likely to confuse many readers not familiar with the details of such analyses. I recommend explaining this term in the supplement where the results of the anlysis are given, and omitting the phrase in the main body of the paper. Suffice to say “Analysis of allelic variants with uncertain clinical signifance (VUS) located in the 5’ non-coding region of GJB2 was performed using Sfold software (https:// etc ).
Line 140: “… with the an area of …”
Line 198: “… identified GJB2 varaints variants, the contribution …”
Line 262: “In the present study, the molecular genetic analysis …” (i.e. omit “the”)
Line 289: Recommend change in wording to: “To our knowledge, our earlier study [42] is the only audiological analysis of a large cohort of patients (N=40) homozygous for the c. -23+1G>A variant. In that study we reported the median PTA 0.5,1.0,2.0,4.0kHz to be 86 dB (severe HI), which is better than that in patients with c.[35delG];[35delG] genotype (102 dB, profound HI)(Figure 4).”
Author Response
Responses to Reviewer 2
Dear Reviewer,
On behalf of the authors, I would like to thank you for your efforts in reviewing our manuscript, and I greatly appreciate your comments.
This is a well-written and well-presented paper.
Thank you very much for your positive feedback about our manuscript.
I suggest a few minor grammatical changes:
(deleted words have strikethough, inserted words are underlined)
We carefully checked our manuscript for typos, errors in spelling, word choice, grammar, punctuation.
Line 60: “… Russia presents the challenges for …” (i.e. omit “the”)
- Fixed
Line 61: “… singificant geographical distances and culturual differences between among numerous …”
- Fixed
Line 86: “ and contact with exposure to industrial noise.”
- Fixed: ..the exposure to industrial noise..
Line 124: The phrase “Ensemble centroid instrument” is likely to confuse many readers not familiar with the details of such analyses. I recommend explaining this term in the supplement where the results of the anlysis are given, and omitting the phrase in the main body of the paper. Suffice to say “Analysis of allelic variants with uncertain clinical signifance (VUS) located in the 5’ non-coding region of GJB2 was performed using Sfold software (https:// etc ).
- We changed this phrase to (Lines: 173-1180): “Analysis of allelic variants with uncertain clinical significance (c.-49G>A and c.-254C>T) located in the 5’ non-coding region of GJB2 was performed using Sfold software (https://sfold.wadsworth.org/cgi-bin/srna.pl) (Supplementary Materials, Chapter 2).
Line 140: “… with the an area of …”
- Fixed
Line 198: “… identified GJB2 varaints variants, the contribution …”
- Fixed
Line 262: “In the present study, the molecular genetic analysis …” (i.e. omit “the”)
- Fixed
Line 289: Recommend change in wording to: “To our knowledge, our earlier study [42] is the only audiological analysis of a large cohort of patients (N=40) homozygous for the c. -23+1G>A variant. In that study we reported the median PTA 0.5,1.0,2.0,4.0kHz to be 86 dB (severe HI), which is better than that in patients with c.[35delG];[35delG] genotype (102 dB, profound HI) (Figure 4).
- Fixed (Lines: 291-295).

Reviewer 3 Report
Review-2023-Genes-2334613-GJB2-Russia-050423
The GJB2 (Cx26) gene variants in patients with hearing impairment in the Baikal lake region (Russia)
By
Pshennikova VG et al
The paper is an extensive study of GJB2 variants and haplotypes in a sample of 165 hearing impaired adults (from 160 families), with average age of 50.7 years. Seventy-nine were of Buryat ethnicity ( 47%) and 76 were Russian (46%, and a few , namely 10 (6%) were of mixed ethnicity. The Republic of Buryatia borders with Mongolia.
The most interesting elements in the paper is that there was no cases with GJB6 deletion, that the exon 1 variant c.(-23+1G) was quite frequent and also found in homozygosity which is extremely rare in the Western countries and USA, and that the fraction of Hearing impairment (HI) in the Buryat patients explained by GJB2 variants was only 5.1%, in contrast to 28.9 % in the Russian patients. This is in agreement with the probably lowest frequency of GJB2 related genetic explanation of hearing impairment in Mongolia. It remains to be studied by means of NGS methodology which genes prevail in Mongolian and related populations.
And it remains to be studied by means of NGS methods which other genes explain the remaining HI cases from both Russia and Buryat people.
All together only 26 out of 165 (they should correct this calculation to only include one proband from each of the 160 families which was explained by GJB2 variants (= 15.8%).
The haplotype analysis supports previous data of founder for the c.(-23+1G) haplotype and the data about 235delC in this paper does not resolve previous discrepancies in origin, where there seems to be differences between Eastern Asia and Northern Asia.
The milder degree of HI associated with the exon 1 variant is well documented in previous papers, albeit homozygosity has been only rarely reported. Conclusions of degree of HI should be more cautious in this sample of adults (average age 50.7 years) since it is unknown to which extent the individuals have noise induced worsening or other environmental additional factors decreasing their hearing ability.
The reading of the paper is complicated by the extensive use of the way results with mentioning of all variants is presented in the paragraph “results”. Several times long listing of all variants are included in the body of the text in many sentences. EVEN ADDITIONAL TABLES MIGHT INCREASE THE READIBILY OF SEVERAL SENTENCES.
Furthermore, a major part of the data could be moved to a supplementary section for people interested in the details of i. e the SNPs and haplotyping.
As the paper stands now it is much too long compared to the novelty of data (which is summarized above).
Author Response
Responses to Reviewer 3
Dear Reviewer,
On behalf of the authors, I would like to thank you for your efforts in reviewing our manuscript. Your constructive comments have allowed us to improve our manuscript. All of your comments have been addressed (please see below), with corresponding changes made directly to the manuscript where appropriate.
Review-2023-Genes-2334613-GJB2-Russia-050423
The GJB2 (Cx26) gene variants in patients with hearing impairment in the Baikal lake region (Russia)
By Pshennikova VG et al
The paper is an extensive study of GJB2 variants and haplotypes in a sample of 165 hearing impaired adults (from 160 families), with average age of 50.7 years. Seventy-nine were of Buryat ethnicity (47%) and 76 were Russian (46%, and a few, namely 10 (6%) were of mixed ethnicity. The Republic of Buryatia borders with Mongolia. The most interesting elements in the paper is that there was no cases with GJB6 deletion, that the exon 1 variant c.(-23+1G) was quite frequent and also found in homozygosity which is extremely rare in the Western countries and USA, and that the fraction of Hearing impairment (HI) in the Buryat patients explained by GJB2 variants was only 5.1%, in contrast to 28.9 % in the Russian patients. This is in agreement with the probably lowest frequency of GJB2 related genetic explanation of hearing impairment in Mongolia. It remains to be studied by means of NGS methodology which genes prevail in Mongolian and related populations. And it remains to be studied by means of NGS methods which other genes explain the remaining HI cases from both Russia and Buryat people.
- Our work is devoted to a comprehensive (molecular-genetic, genotype-phenotype and haplotype analysis) investigation of the GJB2 gene performed for the first time in patients with hearing loss living in the Republic of Buryatia located in the Baikal Lake region (Siberian part of Russia). The study of the GJB2 gene, the most important gene associated with nonsyndromic hearing loss in many populations worldwide, is considered a necessary first step in molecular diagnostics in patients with hearing loss. We agree that further extensive studies (including the NGS technology) are necessary to elucidate the unidentified genetic causes of hearing loss in patients examined in our study. We have added the corresponding phrase to Conclusions (Lines: 362-363):
“Further extensive studies (including the NGS technology) are necessary to elucidate the unidentified genetic causes of hearing loss in patients examined in our study.”
All together only 26 out of 165 (they should correct this calculation to only include one proband from each of the 160 families which was explained by GJB2 variants (= 15.8%).
- The pathogenic contribution of GJB2 variants to the etiology of hearing impairment in the cohort of patients examined in our study was calculated as the proportion of patients with biallelic recessive pathogenic GJB2 variants among ALL patients enrolled (15.8%, 26 out of 165) (Table 2). Whereas, when we estimated the frequencies of GJB2 mutant alleles in two main groups of patients (Buryats and Russians), to exclude possible bias, we used unrelated individuals (Figure 2) and we have specified this issue in the legend of Figure 2 (Lines: 212-213): “The allele frequency of the GJB2 variants was calculated in unrelated patients.”
The haplotype analysis supports previous data of founder for the c.(-23+1G) haplotype and the data about 235delC in this paper does not resolve previous discrepancies in origin, where there seems to be differences between Eastern Asia and Northern Asia.
- All previous studies focusing on the analysis of the haplotypes bearing c.235delC assumed the hypothesis on its single origin. In our study, we found for the first time different haplotypes with c.235delC in Northern Asians that allows speculating about another hypothesis for the origin of the c.235delC mutant allele. We believe our results of haplotype analysis are interesting and relevant because c.235delC is one of the major pathogenic GJB2 variants in many Asian countries.
The milder degree of HI associated with the exon 1 variant is well documented in previous papers, albeit homozygosity has been only rarely reported. Conclusions of degree of HI should be more cautious in this sample of adults (average age 50.7 years) since it is unknown to which extent the individuals have noise induced worsening or other environmental additional factors decreasing their hearing ability.
- We agree that various environmental factors may also influence the degree of hearing loss in older patients. Nevertheless, in our work, we based on the reference genotype c.[35delG];[35delG], for which reliable data on its phenotypic manifestation have already been good documented. Currently, the data on the GJB2 genotype-phenotype correlations in many studies have been obtained on quite heterogeneous samples of patients that often lead to inconsistent conclusions. We believe that our data on the audiological profiles of patients with different GJB2 genotypes will be useful for more deep understanding genotype-phenotype correlations observed for pathogenic variants of this gene.
The reading of the paper is complicated by the extensive use of the way results with mentioning of all variants is presented in the paragraph “results”. Several times long listing of all variants are included in the body of the text in many sentences. EVEN ADDITIONAL TABLES MIGHT INCREASE THE READIBILY OF SEVERAL SENTENCES.
-We believe that the mentioning of all identified variants in the Results section is necessary for the correct presentation of our results. Nevertheless, we removed redundant mentioning of the genotypes in the text of the section “3.2. GJB2 genotypes in patients with HI”.
Furthermore, a major part of the data could be moved to a supplementary section for people interested in the details of i. e the SNPs and haplotyping. As the paper stands now it is much too long compared to the novelty of data (which is summarized above).
- We believe that the haplotype analysis is an important part of our manuscript. However, according to your comment, we slightly shortened the Discussion section by moving Table 3 to the Supplementary materials.

Round 2
Reviewer 3 Report
my previous review is still relevant and valid. the revision is only minor and superficial compared to review comments
Author Response
Responses to Reviewer 3
Dear Reviewer,
On behalf of the authors, I would like to thank you for your efforts in reviewing our manuscript, and I greatly appreciate your comments.
- my previous review is still relevant and valid. the revision is only minor and superficial compared to review comments
- In line with your previous comment “The reading of the paper is complicated by the extensive use of the way results with mentioning of all variants is presented in the paragraph “results. Several times long listing of all variants are included in the body of the text in many sentences…”, we have changed the Results section “3.1. Identified variants in the GJB2 gene” (Lines 157-165) by removing overabundant mentioning of variants and moving the necessary information to Figure 1. We have carefully discussed all your other comments and consider that we have made all possible changes to our manuscript where it was acceptable. We hope that we have adequately addressed all concerns and comments. We believe that our new data about comprehensive (molecular-genetic, genotype-phenotype and haplotype analysis) investigation of the GJB2 gene performed for the first time in patients with hearing loss in the Baikal Lake region (Siberian part of Russia) will be interesting for a wide range of readers, because the GJB2 gene and its major pathogenic variants (c.23+1G>A, c.35delG and c.235delC) is still relevant for more patients with hearing impairment around the world.
